# The Mastery Rubric for Bioinformatics: A tool to support design and evaluation of career-spanning education and training

**Rochelle E. Tractenberg**[1]*, **Jessica M. Lindvall**[2], **Teresa K. Attwood**[3], **Allegra Via**[4]

**1** Collaborative for Research on Outcomes and –Metrics, and Departments of Neurology, Biostatistics, Biomathematics and Bioinformatics, and Rehabilitation Medicine, Georgetown University, Washington, DC, United States of America, **2** National Bioinformatics Infrastructure Sweden (NBIS)/ELIXIR-SE, Science for Life Laboratory (SciLifeLab), Department of Biochemistry and Biophysics, Stockholm University, Stockholm, Sweden, **3** Department of Computer Science, The University of Manchester, Manchester, England, United Kingdom; The GOBLET Foundation, Radboud University, Nijmegen Medical Centre, Nijmegen, The Netherlands, **4** ELIXIR Italy, National Research Council of Italy, Institute of Molecular Biology and Pathology, Rome, Italy

* rochelle.tractenberg@gmail.com

**Data Availability Statement:** All of the data are included in this manuscript: the data are qualitative and are included in the tables.

## Abstract

As the life sciences have become more data intensive, the pressure to incorporate the requisite training into life-science education and training programs has increased. To facilitate curriculum development, various sets of (bio)informatics competencies have been articulated; however, these have proved difficult to implement in practice. Addressing this issue, we have created a curriculum-design and -evaluation tool to support the development of specific Knowledge, Skills and Abilities (KSAs) that reflect the scientific method and promote both bioinformatics practice and the achievement of competencies. Twelve KSAs were extracted via formal analysis, and stages along a developmental trajectory, from uninitiated student to independent practitioner, were identified. Demonstration of each KSA by a performer at each stage was initially described (Performance Level Descriptors, PLDs), evaluated, and revised at an international workshop. This work was subsequently extended and further refined to yield the Mastery Rubric for Bioinformatics (MR-Bi). The MR-Bi was validated by demonstrating alignment between the KSAs and competencies, and its consistency with principles of adult learning. The MR-Bi tool provides a formal framework to support curriculum building, training, and self-directed learning. It prioritizes the development of independence and scientific reasoning, and is structured to allow individuals (regardless of career stage, disciplinary background, or skill level) to locate themselves within the framework. The KSAs and their PLDs promote scientific problem formulation and problem solving, lending the MR-Bi durability and flexibility. With its explicit developmental trajectory, the tool can be used by developing or practicing scientists to direct their (and their team's) acquisition of new, or to deepen existing, bioinformatics KSAs. The MR-Bi is a tool that can contribute to the cultivation of a next generation of bioinformaticians who are able to design reproducible and rigorous research, and to critically analyze results from their own, and others', work.

**Funding:** The author(s) received no specific funding for this work, however, we gratefully acknowledge the Georgetown University Department of Neurology, the GOBLET Foundation, and ELIXIR Italy for defraying the publication costs of this article.

**Competing interests:** Rochelle Tractenberg has read the journal's policy and has the following competing interest: She is a section editor for PLOS ONE; this does not alter our adherence to PLOS ONE policies on sharing data and materials. The other authors have declared that no competing interests exist.

## Introduction

During the past two decades, many commentators [1]; [2]; [3]; [4]; [5]; [6]; [7]; [8]; [9]; [10] have drawn attention to the wide gap between the amount of life-science data being generated and stored, and the level of computational, data-management and analytical skills needed by researchers to be able to process the data and use them to make new discoveries. Bioinformatics, the discipline that evolved to harness computational approaches to manage and analyze life-science data, is inherently multi-disciplinary, and those trained in it therefore need to achieve an integrated understanding of both factual and procedural aspects of its diverse component fields. Education and training programs require purposeful integration of discipline-specific knowledge, perspectives, and habits of mind that can be radically different. This is true whether the instruction is intended to support the use of tools, techniques and methods, or to help develop the next generation of bioinformaticians, and thus can be difficult to achieve, particularly in limited time-frames [11]; [12].

Several international surveys have been conducted to better understand the specific challenges for bioinformatics training (e.g., [13]; [14]; [12]; [15]). While all agree on the necessity of integrating computational skills and analytical thinking into life-science educational programs, they nevertheless acknowledge that the difficulties of achieving this in a systematic and formal way remain. To try to address these issues, various groups around the world began developing curriculum guidelines, defining core (bio)informatics competencies necessary to underpin life-science and biomedical degree programs (e.g., [16]; [17]; [18]; [19]; [20]; [21]). Competencies represent what individuals can do when they bring their Knowledge, Skills and Abilities (KSAs) together appropriately [22], and at the right level(s), for the right application, to achieve a given task. Competencies are thus multi-dimensional, highly complex, task-specific, behaviors.

Using competencies to guide curriculum development has proved problematic (e.g., [23]; [24]; [25]; [26]; [27]). In consequence, the Accreditation Council for Graduate Medical Education in the United States has shifted their definition of "competencies-based medical education" towards a specifically developmental approach that specifies and implements "milestones" [28]; [29]. This is a shift of focus for curriculum design (whether recognized or not) from the end-state of education (i.e., acquired competencies) to *how learners need to change*–to develop–en route to achieving those desired end-states. In other words, curriculum design also needs to include "*the route*" or developmental trajectory [30]. This has been acknowledged in diverse contexts: for medical education, Holmboe et al. (2016) [28] discuss the need to structure "teaching and learning experiences...[in order] to facilitate an explicitly defined progression of ability in stages"; for bioinformatics, Welch et al. (2016) [20] call for community-based efforts to "[i]dentify different levels or phases of competency" and "provide guidance on the evidence required to assess whether someone has acquired each competency." In other words, both the steps along the way *and the way* are requisite to a curriculum that supports the achievement of milestones and competencies (see [31]).

In developing curricula that both promote acquisition, integration, and retention of skills *and* incorporate developmental considerations, part of the challenge is the time-frame available for instruction: formal education can follow structured, long-term programs, but *training* is generally delivered within limited time-frames [11]; [12]. In such circumstances, the time available to align instruction with learners' levels of complexity of thinking (and experience) is limited; other considerations, including the prior experience and preparation of the learners, are also factors. Nevertheless, it is possible to address developmental considerations for both short- and long-form learning experiences [32] by appeal to Bloom's Taxonomy of the Cognitive Domain [33], which specifies a six-level hierarchy of cognitive skills or functioning:

1. *Remember/Reiterate*—performance is based on recognition of a seen example(s);

2. *Understand/Summarize*—performance summarizes information already known/given;

3. *Apply/Illustrate*—performance extrapolates from seen examples to new ones by applying rules;

4. *Analyze/Predict*—performance requires analysis and prediction, using rules;

5. *Create/ Synthesize*–performance yields something innovative and novel, creating, describing and justifying something new from existing things/ideas;

6. *Evaluate/Compare/Judge* [34]—performance involves applying guidelines, not rules, and can involve subtle differences arising from comparison or evaluation of abstract, theoretical or otherwise not-rule-based decisions, ideas or materials. (This representation, with "evaluate/judge" at the pinnacle, is from the original Bloom taxonomy, while the 2001 revision characterized "create/synthesize" as the most cognitively complex ([34]; see [30], for discussion of how the original formulation suits the higher/graduate/post graduate context)).

The hierarchy in Bloom's taxonomy is developmental [30]; this highlights a problem for some of the proposed competencies for practicing scientists (e.g., [17]; [19]). Specifically, most of the articulated competencies require very high-level Bloom's, and nearly all require the use of *several* Bloom's cognitive processes seamlessly, sometimes iteratively. Without a specific built-in developmental trajectory that can lead a learner from lower- to higher-level Bloom's (and thence to the integration of the multiple cognitive processes on the spectrum of activities that bio-/medical informatics practice requires), the competencies may simply be too cognitively complex to serve as achievable end-states.

## The Mastery Rubric: A curriculum-development and -evaluation tool

Given the challenges, it is perhaps not surprising that efforts to incorporate competencies into teaching and training have been problematic. Motivated by these issues, we have created a new curriculum-development and -evaluation tool: the Mastery Rubric for Bioinformatics (MR-Bi). Rubrics are typically used to provide a flexible but rigorous structure for evaluating student work [35]; a Mastery Rubric is similar but describes the entire curriculum rather than individual assignments [30]. Creating a Mastery Rubric requires three key steps: 1) identifying the KSAs that a curriculum should deliver, or that are the targets of learning; 2) identifying recognizable stages for the KSAs in a clear developmental trajectory that learners and instructors can identify, and that instructors can target in their teaching and assessment; and 3) observable Performance Level Descriptors (PLDs; [36]) on each KSA at each stage, describing evaluable changes in performance from less to more expert. Within a Mastery Rubric, PLDs clarify what instructors need to teach *and assess* at each stage, and articulate to students what they need to demonstrate, in order for the KSAs to be characterized as 'achieved' for that stage. To date, the Mastery Rubric construct has been used to design and evaluate graduate and postgraduate/ professional curricula in clinical research, ethical reasoning, evidence-based medicine, and statistical literacy (see [30]).

Like competencies, KSAs can be highly complex; by contrast, however, KSAs are general–i.e., the same KSA can be deployed differently to support different task-specific competencies. Hence, this paper focuses on KSA-based teaching and learning, to promote the likelihood of learners' adaptability to future new competencies. We emphasize fostering the *development* of KSAs within a structured framework that explicitly supports continuing growth and achievement. Specifically, we present the MR-Bi, a tool created to support the articulation and

demonstration of actionable teaching and learning goals. The focus of this paper is on how the tool was developed; its uses and applications are the topics of our ongoing work.

## Methods

Construction of the MR-Bi followed formal methods. The KSAs were derived via cognitive task analysis [37]; the stages were derived using Bloom's taxonomy and the European guild structure ([38]; p.182), which maps how individuals can expect/be expected to grow and develop; and the PLDs evolved through a formal standard-setting procedure [39], considering key aspects of the kind of critical thinking necessary in a given discipline. This application of standard-setting methodology (for a review, see [40]) focused on qualitatively, rather than quantitatively, characterizing the performance ("body of work") of the minimally competent performer of each KSA at each stage. The interested reader is encouraged to review the work-flow presented in the Supplemental Materials (Figure A in S1 File). This comprises a detailed description of the integration of theory, the cognitive task analysis, and the drafting of the KSAs using Bloom's taxonomy and the Messick criteria, with examples.

Face and content validity of the MR-Bi were also determined systematically. A degrees of freedom analysis [41]; [42] was used to formally explore the alignment of the KSAs with existing biomedical informatics and bioinformatics competencies ([17], [19]). Alignment of the MR-Bi with principles of andragogy ([43]) was also investigated. These alignment efforts serve to demonstrate whether and how the MR-Bi is consistent both with the goal of supporting achievement of competencies, and with andragogical objectives. The KSAs must be teachable to adults, and the PLDs must describe observable behaviors, if the MR-Bi is to be an effective and *valid* tool for instructors and learners.

### KSA derivation via cognitive task analysis

A cognitive task analysis generated KSAs following the procedure described in the Supplemental Materials (Table A, Figure A, Text A in S1 File). KSAs that characterize the scientific method, reflecting what is requisite in scientific work (derived from [44] and [45], and adapted by [46]), were assumed to be essential to bioinformatics education and training [47]. These KSAs were refined with respect to community-derived competencies. The Welch et al. [19] competencies focus on the development of the *bioinformatics* workforce, while those of Kuli-kowski et al. [17] focus explicitly on curriculum development for doctoral training in *health and medical informatics*. Despite considerable overlap between them, we reviewed both sets to ensure that our cognitive task analysis, and the resulting KSAs, were comprehensive.

### Identification of stages

In a Mastery Rubric, KSAs are described across a developmental continuum of increasing complexity. The specific developmental stages, derived from the European guild structure, are Novice, Beginner, Apprentice and Journeyman ([30]; [48]; see also [49] for a recent similar strategy). In the MR-Bi, these stages use Bloom's taxonomy explicitly to characterize the interactions of the individual with scientific knowledge (and its falsifiability). Generally, someone who deals with *facts* (remembering them, but not questioning them or creating novel situations in which to discover them) is a Novice. A Beginner has a growing understanding of the experimental origins of facts they memorized as a Novice. The individual who can participate in experiments that are designed for them, making predictions according to rules they have learned, but not interpreting or evaluating, is an Apprentice. Although they are not explicit about this, undergraduate life-science programs generally support the transitions from Novice

to Apprentice. These stages characterize the preparatory phases of anyone new to bioinformatics, irrespective of age or prior experience/training.

The guild structure characterizes the independent practitioner as a Journeyman: postgraduate education, or equivalent work experience, supports the transformation from Apprentice (learning the craft) to Journeyman (practitioner). Journeyman-level individuals are prepared for independent practice in the field, although newly independent practitioners generally still require some level of mentorship (e.g., in a post-doctoral context); such individuals are designated *J1 Journeyman*. The scientist whose doctoral program, or background and experience, has prepared him/her for fully independent scientific work is the *J2 Journeyman*. Crucially, an individual with a PhD (or equivalent) in biology, computer science or other scientific field may be a novice in *bioinformatics*: e.g., someone who deals with facts about programming or data resources, but who is unable to apply them to novel situations to discover new biological knowledge (which would require the higher-order cognitive functioning characteristic of the J1 Journeyman or J2 Journeyman performer). These "high level" descriptions guided the PLD-drafting process.

The developmental trajectory in a Mastery Rubric is evidence-based, *not time-based*: the performance of a KSA at any stage should lead to concrete and observable output or work products that can be assessed for their consistency with the PLDs by objective evaluators. Claims of achievement cannot be based on age, time-since-degree, job title, time-in-position, or other time-based indicators.

## Standard setting for PLDs of each KSA at each stage

We followed the Body of Work approach [50] to writing the PLDs, refining descriptions of how a "minimally competent" individual [51] would carry out the KSAs at each stage to demonstrate that they were capable of performing them at that level. This standard-setting exercise (which commenced at a 2-day international workshop in Stockholm in September 2017) was intended to describe performance at the "conceptual boundary between acceptable and unacceptable levels of achievement" ([52], p. 433). Participants were bioinformatics experts working in the National Bioinformatics Infrastructure Sweden (https://nbis.se/). All had documented expertise within the field, ranging from programming to applied bioinformatics with a biological and/or medical focus; all were also highly engaged in training activities within the life science community across Sweden. Prior to the workshop, a white paper on the Mastery Rubric with draft KSAs and PLDs, and the Kulikowski et al. and Welch et al. competencies were sent for viewing and preparation. These and the background and methods outlined here were reviewed during the first half of day 1.

The workshop aimed to accomplish two facets of the PLDs: *range finding* and *pin-pointing* ([50]. P. 202–203). Range-finding involved writing relatively broad descriptions of performance for each KSA at each developmental stage. To orient the participants, pin-pointing initially involved whole-group evaluation and revision of the PLDs across all stages for just three of the KSAs. Afterwards, participants were divided into three small, facilitated groups (n = 4 per group); each undertook pin-pointing of the draft PLDs for four KSAs during the remainder of the two days. Participants iteratively evaluated the KSAs and PLDs to ensure that they and the stages they represent made sense, and that the PLDs were plausible, consistent *within each stage* and not redundant across KSAs. A particular focus was on performance levels that characterize the independent bioinformatician; specifically, to determine whether the Journeyman level for each KSA was realistically achieved at one point (J1 Journeyman), or whether a second stage of development was required (J2 Journeyman) to achieve "full" independence. Thus, all PLD drafting and review/revisions were systematic and formal, following Egan et al.

(2012 [36], pp 91–92) and Kinston & Tiemann (2012 [50], pp 202–203), grounded both on expectations for earlier achievement and performance, and on how an individual could be expected to function once any given stage had been achieved.

### Integrating KSAs, stages, and standards into the MR-Bi

The tasks performed by subject experts during the Stockholm workshop resulted in a completed first draft of the MR-Bi. Afterwards, final refinements (further pin-pointing and range-finding) were made during weekly online meetings (2017–2019). Here, our effort was directed at revising the PLDs to build consistently across stages *within a KSA*, and to describe performance within stages *across KSAs*. The intention was that the PLDs should support conclusions about "what is needed" as evidence that a person has achieved a given stage for any KSA. To this end, the PLDs were aligned with Bloom's taxonomy, and written to reflect the core aspects of assessment validity outlined by Messick (1994 [53]):

1. What is/are the KSAs that learners should possess at the end of the curriculum?

2. What actions/behaviors by the learners will reveal these KSAs?

3. What tasks will elicit these specific actions or behaviors?

Application of the Messick criteria ensured that the PLDs would represent concrete and observable behaviors that can develop over time and with practice. In particular, we focused on Messick questions 2 and 3 so that the MR-Bi would describe specific actions/behaviors learners or instructors could recognize as demonstrating that the KSA had been acquired to minimally qualify for a given stage. Overall, the creation of the MR-Bi followed methods intended to yield a psychometrically valid tool (e.g., [54]).

## Results

### KSA derivation via cognitive task analysis

There are eight KSAs that characterize the scientific method (define a problem based on a critical review of existing knowledge; hypothesis generation; experimental design; identify data that are relevant to the problem; identify and use appropriate analytical methods; interpretation of results/output; draw and contextualize conclusions; communication). These were customized and enriched based on consideration of the 45 competencies articulated by Kulikowski et al. (2012) [17] and Welch et al. (2014) [19], and ultimately, 11 distinct KSAs were derived.

Both sets of competencies include at least one about ethical practice or its constituents, but neither was sufficiently concrete to support a separate KSA. The PLD-writing process therefore began with the intention of integrating features of ethical science (e.g., promoting reproducibility, emphasizing rigorous science and a positivist scientific approach, and specifying attention to transparency) wherever these are relevant. However, it became clear that respectful practice, and awareness of the features of ethical conduct and misconduct, were missing. These considerations led to the identification of a 12th KSA, "ethical practice" (and the range-finding and pinpointing exercises were carried out again–see Supplemental Materials, Figure A in S1 File):

1. Prerequisite knowledge—biology

2. Prerequisite knowledge—computational methods

3. Interdisciplinary integration

4. Define a problem based on a critical review of existing knowledge

5. Hypothesis generation

6. Experimental design

7. Identify data that are relevant to the problem

8. Identify and use appropriate analytical methods

9. Interpretation of results/output

10. Draw and contextualize conclusions

11. Communication

12. Ethical Practice

The KSAs are easily recognizable as key features of bioinformatics practice. They are not intended to be restrictively factual (i.e., literally just knowledge), nor are they content-specific, because content is apt to change quickly and often: this makes the Mastery Rubric both durable and flexible with respect to discipline, obviating the need to develop a different Mastery Rubric for every sub-discipline. Hence, only two types of *Prerequisite knowledge* were identified: in *biology* and *computational methods*. A distinct KSA for *Interdisciplinary integration* was identified as a separate need, because developing the ability to integrate across those domains, and understanding the potential need for inclusion of other domains (biomedicine, statistics, engineering, etc.), are essential in bioinformatics.

*Defining a problem based on a critical review of existing knowledge* underpins the application of critical evaluation skills and judgment, based on what is already known, to determine what is *not yet known*. This KSA supports the definition of a bioinformatician as a scientist who uses computational resources to address fundamental questions in biology [55]; [56]–i.e., it is intended to promote the bioinformatician's ability to *solve biological problems*. It also derives implicitly from the competencies, and explicitly from the Wild & Pfannkuch (1999) [44] model of scientific reasoning (and highlighted in [46]). *Hypothesis generation* also emerged from the cognitive task analysis by appeal to theoretical and empirical scientific-reasoning models.

*Experimental design*, a crucial aspect of the scientific method, was included as a separate KSA in order to cover formal statistics, hypothesis testing, methodological considerations and pilot/sensitivity testing; this allowed the *Prerequisite knowledge* KSAs for *biology* and *computational methods* to focus on the background knowledge, and basic skills and abilities, that are foundational in each area, and the role of experiments and troubleshooting in each.

Given some level of overlap between them, consideration was given to whether KSAs needed to be separate. For example, we discussed whether statistical and engineering methods required their own KSA. The decision to include aspects of statistical inference and experimental design in each of the *Prerequisite knowledge* KSAs was based on a (perhaps aspirational) objective to define these basic features as "prerequisite". This formulation left the more complex and interdisciplinary characteristics of experimental design to its own KSA. Engineering was deemed to be an optional domain, not essential to the specific KSAs required for solving biological problems.

*Identify data that are relevant to the problem* and *Identify and use appropriate analytical methods* were deemed sufficiently distinct to warrant separate KSAs. Similarly, *Communication* was included separately to emphasize the importance of being able to transparently write about and present scientific work, even though there are requirements for communication in

several other KSAs, including *Interpretation of results/output* and *Draw and contextualize conclusions*. Finally, we recognized *Ethical practice* as a separate KSA, in spite of the inclusion of key attributes of ethical science in all of the PLDs relating to transparency, rigor, and reproducibility.

## Identification of stages

Given these KSAs, and Bloom's taxonomy, stages on the developmental trajectory were articulated as follows:

**Novice (e.g., early undergraduate/new to bioinformatics), Bloom's 1**: remember, understand. Novices can engage with well-defined problems, with known solutions.

**Beginner (e.g., late undergraduate, early Master's), Bloom's 2–3**: understand and apply. Beginners may use, but not choose, tools; they can engage with well-defined problems *and apply what they are told to apply*; the answers may not be known, and the Beginner would stop once this was apparent.

**Apprentice (e.g., Master's, early doctoral student), Bloom's 3–4, early 5**: *choose and apply* techniques to problems that have been defined (either jointly or by others). The Apprentice can analyze and interpret appropriate data, identify basic limitations, conceptualize a need for next steps, and contextualize results with extant literature.

**Early Journeyman (J1) (e.g., late doctoral student or just after graduation), Bloom's 5, early 6**: begin to evaluate (review) and synthesize novel life-science knowledge, and to develop abilities to integrate bioinformatics into research practice, with some mentorship. The J1 Journeyman can contribute to problem formulation, shows earliest establishment of independent expertise in the specific life-science area, and can confidently integrate current bioinformatics technology into that area.

**Late/advanced Journeyman (J2) (e.g., doctorate holder), Bloom's 5, late 6**: expertly evaluate (review) and synthesize novel life-science knowledge, and integrate bioinformatics into research practice. The J2 Journeyman is independent and expert in a specific life-science area, and can select, apply and develop new methods. The J2 Journeyman formulates problems, considers the relevance of "what works" within this area *to other* life-science domains, so as to be an adaptable and creative scientific innovator without having to reinvent every wheel.

As already noted, PLD reviewers were instructed to determine whether two different developmental stages should be described for the independent bioinformatician on each of the KSAs. All KSAs were judged to require the two Journeyman levels.

## Synthesizing KSAs, stages, and PLDs into the MR-Bi

As described earlier, 12 KSAs were defined, and PLDs were drafted and iteratively refined for each of the stages on the developmental trajectory. Refinements included, for example, ensuring that neither KSAs nor PLDs included specific tools, programs, or tasks (e.g., BLAST, Hadoop, creating GitHub repositories), because these may change over time but the KSAs will change less quickly, if at all. The results are shown in the MR-Bi in Table 1.

In addition to the KSAs and PLDs, considerations for "evidence of performance" were incorporated into the MR-Bi (Table 1). As noted, the Mastery Rubric is a true rubric in the sense that learners can demonstrate their achievement of any given level on any KSA using a wide variety of evidence. The development of independence as a learner, and as a practitioner, requires a level of self-assessment that is not often considered in higher education and training.

**Table 1. Mastery Rubric for Bioinformatics (MR-Bi).**

| Performance Level: | Novice | Beginner | Apprentice | J1 Journeyman | J2 Journeyman |
|---|---|---|---|---|---|
| **General description of bioinformatics practitioner** | Reads, generally understands, but does not question, life science research (results). Beginning to recognize that "facts" are actually just the best-currently-supported theory. Limited engagement with uncertainty associated with "facts"; developing understanding of experimental design paradigms in biology, & own specific area of study. | Consolidates reading & understanding, beginning to learn how to analyze given biology problems (with software). Growing recognition that "facts" are typically the best-currently-supported theory. Engaging consistently with uncertainty associated with "facts"; deepening understanding of experimental design paradigms in biology, & own specific area of study. | Reads & understands; reliably identifies methods (software & programming) for given problems. Chooses & executes correct analysis, not necessarily able to identify several methods that could be equally viable, depending on given research objectives. Qualified as a fluent, but not as an independent, scientist who uses bioinformatics as a tool, but does not yet synthesize technology with biology to generate new research problems. | Qualified as an independent scientist who uses bioinformatics methodologies as part of routine practice. Poses novel scientific questions, & identifies data & technology to align appropriate statistical/analytical methods to desired scientific objectives. Experienced reviewer of relevant technical features of available bioinformatics methods. Newly-independent expert in integrating bioinformatics technology/ techniques into novel research problems in their area of expertise. | Independent scientist who expertly integrates bioinformatics & more traditional methodologies, as needed, to achieve desired objectives & contribute to the body of knowledge. Expert reviewer of relevant technical features of available bioinformatics options. |
| **Considerations for evidence of performance at this level** | **Bloom's 1, early 2**: remember, understand. Problems the Novice can engage with are well-defined, with solutions already known. Work does not generally reflect self-assessment. | **Bloom's 2–3**: understand & apply, **but only what they are told to apply**. Problems the Beginner can engage with are well-defined. Work reflects some self-assessment, when directed to do so. | **Bloom's 3–4, early 5**: *choose & apply* techniques to problems that have been defined (either jointly or by others). Can analyze & interpret appropriate data, identify basic limitations & conceptualize a need for next steps/ contextualization of results with extant literature. Seeks guidance to improve self-assessment of own work. | **Bloom's 5, early 6**: evaluate (review) & synthesize novel life-science knowledge while developing abilities to integrate bioinformatics into research practice. Shows independent expertise in a specific life-science area, & confidently integrates current bioinformatics technology into that area. Beginning to critically evaluate experimental paradigms & their results, without knowing/ requiring that there be "one right answer". Consistently self-assesses own work. | **Bloom's 6**: prepared for independent scientific work. Expert in design & critical evaluation of experimental paradigms & their results. Self-assesses in own work, & encourages others to develop this skill. |
| **Ethical practice** | Exhibits respect for community standards/rules for public behavior & personal interaction. Learning how to recognize, & manifest respect for, intellectual property, professional accountability, & scientific contributions. | Learning to recognize "misconduct" in the scientific sense. Learning to avoid, & respond to, misconduct; & the importance of neither condoning nor promoting it. | Learning the principles of ethical professional & scientific conduct. Seeks guidance to strengthen applications of these principles in own practice. Learning how to respond to unethical practice. | Practices bioinformatics in an ethical way, & does not promote or tolerate any type of professional or scientific misconduct. Seeks guidance in how/when to take appropriate action when aware of unethical practices by others. | Practices, & encourages all others to practice, bioinformatics in an ethical way. Does not promote or tolerate any type of professional or scientific misconduct. Takes appropriate action when aware of unethical practices by others. |
| **Prerequisite knowledge– biology (includes statistical inference & experimental design considerations)** | Basic knowledge of biology; little-to-no awareness of the uncertainty inherent in experimental designs common in the life sciences. Thinking about the life sciences is based on uncritical acceptance of information as "factual" or "true". | Advanced knowledge of biology, & basic knowledge of key bioinformatics methods. Very simple statistics/programs are run to answer pre-defined scientific questions. Learning to understand the uncertainty inherent in the scientific method, questions assumptions in the data & their relevance for given scientific problems (which arise from others). | Thinking about life sciences integrates both experimental & bioinformatics/ technological sources for data & knowledge. Understands the uncertainty inherent in the scientific method, questions assumptions in the data & their relevance for given scientific problems (which typically arise from others, or with others). Experimental design & statistical inference are recognized & exploited with guidance, to answer given scientific problems. Can recognize inconsistencies in biological data/ experiments that are identified by others, but cannot troubleshoot experimental methods independently. | Recognizes the importance of, & is able to critically evaluate, the relevant literature, & understands historical background of the relevant biological system(s). Sufficient knowledge of a biological system(s) to be able to draw functional conclusions from analytical results. Collaborates with experts to inform the next stages in the experimental design process (validating results, follow-up analyses, etc.). | Makes predictions to inform next stages of experimental design process. Evaluates relevant experimental methods that can be applied in any problem. Can generalize to other biological systems; independently solves biological problems that are innovative & move the field forward. |
| **Prerequisite knowledge– computational methods (includes statistical inference & experimental design considerations)** | Basic knowledge of computational methods; little-to-no awareness of the relevance of computational methods for life sciences. No awareness of experimental designs or how these can be used or implemented in computational applications Thinking about tools, computers, software, & programming is strictly uni-dimensional: i.e., extrapolation &/or abstraction of knowledge about computational methods to other systems, programs, or problems, are not possible. Can run software or execute code they are given (as appropriate) with precise instructions; cannot write a script or debug/troubleshoot. | Computers, software, tools, & programming are understood to be options for scientific work. Learning how to write & test code, run software, or use tools, as appropriate. Is developing awareness of the variety of bioinformatics tools, designs, & resources, but is not able to choose or apply the most appropriate of these for any given question; when choices are made, tools are used uncritically. Developing awareness that computational tools require input parameters, but uses the default settings. Learning to read, understand, troubleshoot, & make minor modifications to existing code/scripts. Does not synthesize results or outputs. | Learning to test software & programming approaches to different types of problem. Experimental design & statistical inference using computing & algorithms are recognized & applied, with guidance, to answer given scientific problems. Learning "best practices" for programming, if programming is part of the task. Can write basic code in a given language or run appropriate software, using judgement, but not inventing or innovating. Cannot troubleshoot complex computational methods–will ask for guidance. Exploring alternatives to default input parameters across computational tools. Can apply knowledge of tools to interpret their results & output. Seeks guidance in synthesis of results or outputs. | Recognizes the importance of, is able to critically evaluate, & understands historical background of the relevant data, databases, tools, data analysis/ statistical methods & computational resources. Can utilise these & justify trade-offs across methodologies (e.g., which statistical test to apply & what computational methods to use). Collaboratively synthesizes & critically questions analysis results & output from tools. Recognises the iterative nature of experiments (e.g., bench, data analysis, back to bench). Can write code/use tools to accomplish these, but collaborates with domain experts for identifying & articulating biological problems that are innovative & move the field forward. | Develops robust, well-documented, optimized, reproducible code &/or uses tools to address biological problems; moves away from standard procedures & innovates to accommodate new data types, tools, & techniques as needed. Can generalize to new coding languages or software/tools/ resources. |

*(Continued)*

**Table 1.** (*Continued*)

| Performance Level: | Novice | Beginner | Apprentice | J1 Journeyman | J2 Journeyman |
|---|---|---|---|---|---|
| **Integrate interdisciplinarity** | Does not recognize life sciences as requiring integration of both experimental & computational/modeling approaches. Perceives disciplines as separate; integration only occurs when/as directed. Information, ideas & tools that are inter-disciplinary are used without question. | Beginning to think about life sciences as requiring integration of both experimental & computational/modeling approaches. Recognizes that interdisciplinarity is needed, but does not know how (or when) to do it, & requires direction. Learning the integrating process; learning strengths & weaknesses of biological & computational methods, but not sufficient to question assumptions from these & other disciplines. | Understands that life sciences integrate both experimental & computational/modeling approaches; seeks guidance about how & when to integrate. Developing an understanding of the strengths & weaknesses of biological & computational methods, beginning to question fundamental assumptions from these & other disciplines for any given scientific problem (which typically arises from others, or in conjunction with others). | Collaboratively integrates across relevant disciplines to address, & solve, innovative biological problems. Tests multiple avenues to triangulate solutions, with minimal guidance. Recognizes the roles of interdisciplinary teams in the research process, & the importance of integrating interdisciplinarity early on. Works effectively on interdisciplinary teams with minimal guidance. | Formulates innovative biological problems that require interdisciplinary solutions. Integrates methods & results to derive & contextualize solutions to biological problems. Consistently tests multiple avenues to triangluate solutions, while exploiting relevant findings from other disciplines. Actively builds interdisciplinary teams, as needed. |
| **Define a problem based on a critical review of existing knowledge** | Can recognize a problem that is explicitly articulated or concretely given, but cannot derive one. Unaware of the depth & breadth of "the knowledge base"that is or could be relevant for the formulation of a problem. Does not recognize design features or other evidence as the basis of/support for problem articulation. Does not recognize uncertainty or how this affects the formulation of solveable problems. | Developing awareness of the depth & breadth of "the knowledge base"that is or could be relevant for the formulation of a problem. Cannot differentiate gaps in own knowledge from gaps in "the knowledge base". Developing the ability to recognize that uncertainty may have arisen in the formulation of solutions to problems. | Beginning to use, with guidance, the appropriate knowledge base to address a given problem. Recognizes the need to consider a wider scope of knowledge for alternative solutions to a problem common across contexts or domains. In guided critical reviews, learning to recognize that design features & evidence base are important to drawing conclusions. Recognizes the role of uncertainty in research, & that reproducibility & potential bias should be considered for every result. | Can explore & critically review the relevant knowledge base, & collaboratively articulate a problem based on that review. Reviews include assessment of relevance from (potentially) ancillary domains, bias, reproducibility, & rigor; recognizes when appropriate & inappropriate methodology is used. Recognizes when incomplete review is provided (by themselves or by others). Can discern reproducible from non-reproducible results; can identify major sources of bias throughout the knowledge base. | Independently defines & articulates a theoretical or methodological problems based on a critical review of the relevant knowledge base(s). Knows the hallmarks of questionable research hypotheses & misalignment of testing/statistics with poorly articulated research problems; consistently finds & identifies sources of bias. Articulates when appropriate & inappropriate methodology is used/reported. Critical review & problem articulation integrate diverse disciplinary perspectives when appropriate/adaptable. |
| **Hypothesis generation** | When directed, follows instructions to *test* hypotheses; does not generate them & may not recognize them without explication. Uses the default settings of software & other tools, rather than a hypothesis, to guide any analysis. Does not question methods to be used, or assumptions of methods that are used. | When directed, uses the default settings of software, tools, or the GUI to test hypotheses in pre-planned analyses; does not generate testable hypotheses. Does not recognize that hypotheses may be generated & tested within the intermediate steps of an analysis. Developing the understanding that all methods involve assumptions. | With guidance, can: 1. leverage tools, software, data & other technologies (GUI/programming) to test hypotheses; & 2. generate hypotheses based on either the data or the technology, but not their combination/synthesis. Hypothesis generation possible in highly concrete & fully parameterized problems; developing the ability to identify whether a given hypothesis -including one of their own—is testable. Learning to recognize that experimental design & design of software/programming solutions include hypothesis generation to some extent. Developing the abilities to identify, & plan to address, assumptions that different hypotheses necessitate. | Collaboratively integrates hypothesis generation into the consideration of literature, data & analysis options. Seeks appropriate guidance in the synthesis of data & technology to generate novel, testable hypotheses. Considers the process of hypothesis generation & testing to be iterative when this is appropriate. Hypothesis generation is done with consideration of reproducibility & potential for bias, & takes into account the most clearly relevant literature; recognizes that less-obviously relevant literature may also be informative for hypothesis generation. | Independently generates testable hypotheses that are scientifically innovative as well as feasible (possible for economic reasons, time, impact, etc.). In own & others' work, recognizes that, & articulates how, hypothesis generation from planned & unplanned analyses differ in their evidentiary weight & their need for independent replication. Fully explores all relevant knowledge base(s) to support rigor & reproducibility, & to avoid bias, in the generation of hypotheses. |
| **Experimental Design** | Can recognize concrete features of experiments only if they are described/given, and they match basic design elements (e.g., dependent, independent variables). Cannot design data collection or experiments. Unaware of covariates or their importance in analysis or interpretation. Does not recognize the importance of design, data collection, data quality, storage/access, analysis, & interpretation to promote rigor & reproducibility in experimental design. | Can identify salient features of experiments that are described/given if they match previously encountered design elements, but cannot derive them if they are not present. Recognizes covariates if mentioned, but does not require formal consideration (or justification) or evaluation of covariates. Does not recognize that one experiment alone cannot adequately address meaningful biological research problems. Understands that experimental design involves the identification, gathering, storing, analyzing, interpreting, & integrating of data & results. | Can match the correct data collection design to the instruments & outcomes of interest. May include/exclude covariates, or other design features, "because that is what is done", without being able to justify their roles in the hypotheses to be tested. Developing the understanding that weak experimental design yields weak data & weak results. Needs assistance in conceptualizing covariates & their potential roles in the planned analyses. Beginning to recognize that, & can explain why, just one study is usually insufficient to answer a given research problems/solve biological problems adequately. Follows templates for the identification, gathering, storing, analyzing, interpreting & integrating of data. Learning to consider reproducibility & rigor in experimental design, & to question templates that do/do not include these concepts. | Recognizing that explicit attention to experimental design will result in more informative data; designs experiments in consultation with experts in content & statistics. These experiments may include power calculation considerations, if relevant; modeling requirements; measurement/sampling error & missing data. Collaboratively designs experiments that address the need for reproducibility & sensitivity analysis. Learning to conceptualize pilot studies & sensitivity analyses. Learning to adapt problems so that hypotheses can be generated & made testable via experiments. | Independently designs appropriate & reproducible experiments & other data-collection projects, using methodologies that are aligned with the testing of specific hypotheses. Consistently identifies & justifies choices of instruments & outcomes (& covariates if relevant). Collaborates with experts as needed on appropriate use of advanced methods, including accommodating measurement & sampling error, attrition (if needed) & modeling requirements; can adapt complex problems so that hypotheses can be generated & made testable via experiments. Understands & can exploit the strengths & weaknesses of experimental design, data & modeling approaches with respect to the biological problem under consideration. Uses pilot studies & sensitivity analyses appropriately. |

(*Continued*)

**Table 1.** (Continued)

| Performance Level: | Novice | Beginner | Apprentice | J1 Journeyman | J2 Journeyman |
|---|---|---|---|---|---|
| **Identify data that are relevant to the problem** | Uses data, as directed. Does not find relevant data; cannot describe what makes data or a given data resource "relevant" to a given problem. | Correctly uses data that are provided or can follow a script/"recipe" to obtain (access, manage) relevant data to which they are guided. Cannot determine whether a given data-set or type is relevant for a given problem, but is developing an awareness that not all data are equally relevant, or equally well suited, to all research problems. Developing awareness of the features of data/data resources that constitute "relevance", & that these features must be assessed before choosing data to use. | Can initiate a search for data & will ask if uncertain about the relevance for any given problem. Learning how to identify, & evaluate strengths & weaknesses of, data resources, to determine whether a given data-set or -type is relevant for a given problem; &, with guidance, how to leverage these to address given research problems. Learning how reproducibility can be affected by the choice (& features) of data. | Collaboratively identifies relevant data resources. Understands the relative strengths & weaknesses of data-sets & -types for addressing their specific problem. Learning to address & formulate scientific problems (based on recognized gaps in the knowledge base) utilizing relevant data resources. In own & others' work, recognizes that, & articulates how, choices for data (collection or use) require assumptions & justification, & must yield reproducible results. | Identifies data that are directly relevant to a problem of own or others' devising. Consistently identifies, & evaluates strengths & weaknesses of, a variety of data resources that can address a problem or help to formulate it more clearly; recognizes if the necessary data do not yet exist. Justifies the relevance of any given data-set to a problem in terms of their individual strengths & weaknesses. Articulates hypotheses, & designs experiments, that leverage strengths in the data; includes triangulating data or results to address weaknesses in the data. Identifies whether data appropriate to the specific scientific question were used when reviewing proposals, protocols, manuscripts, &/or other documentation describing data, & research results. |
| **Identify & use appropriate analytical methods** | Uses methods that are provided & in a given order (i.e., a pipeline; & treats workflows* as if they are pipelines). Does not identify relevant methods; cannot describe what makes a method "relevant" to a given problem. Unaware that methods & software have default settings. Does not question propriety, assumptions, or order of methods that are employed; focus is on the superficial attributes of given methods & protocols. | Uses given methods, as directed, & learning about the concepts of pipelines & workflows*; still uses workflows as if they are pipelines, but beginning to attend to decision points. Learning to recognize pros & cons of methods/software, but cannot yet effectively compare, evaluate, or rank them. Becoming aware of the default settings of software or methods & their effects on results; & beginning to explore & inquire about tailored settings. Understands that more than one method/software may be available to deal with a given problem or data type, but can't choose effectively. Learning about similarities & differences across methods, & that choices (particularly of multiple methodologies for one question) should leverage independence of methods to support reproducible results. | Can identify methods, software, & pipelines that are relevant for a given problem; seeks guidance about the best approach. Learning to evaluate/rank & justify alternative methods in terms of general features of their efficiency & relevance for the given research problem. Beginning to recognize that a "pipeline" involves only the choice of which one(s) to use; while a "workflow" requires many choices & decisions. With guidance, seeks to identify & implement appropriate workflows to address given research problems. Learning how reproducibility can be affected by the choice & implementation of methods, including independent replication of essentially the same method vs. independent replication using diverse methods. | Collaboratively considers the knowledge base, & features of the relevant data & analysis options, in identifying the most appropriate approach(es) to tackle a scientific question. Uses appropriate analytic methods, pipelines, & workflows, recognizing, & taking advantage of the fact, that these may represent distinct approaches to the same problem. Knows when & how to control false discovery rates to promote reproducible results across methods. In own & others' work, recognizes that, & articulates how, choices for methods, pipelines, & workflows require assumptions & justification, & must yield reproducible results. | Recognizes if/when the necessary methods, pipelines, & workflows to tackle a scientific question do *not yet* exist. Consistently controls false discovery rates to promote reproducible results. Identifies whether appropriate analytical methods were used when reviewing proposals, protocols, manuscripts, &/or other documentation describing methods, pipelines, workflows, & research results. |
| **Interpretation of results/ output** | Treats the output of a program as the final/complete result–with no interpretation required–& is unaware of the concepts of validation & cross-validation or their importance for interpretation of results/output. Uses the p-value to indicate "truth" in statistical analysis. Over-interpretation is typical. Unaware of the importance of false discovery rate controls. Does not seek coherence in/recognize incoherence of their results with the analysis plan or pipeline; is unable to align methods, results, & interpretation. | Interpretation of results depends on *p*-values, but understanding of *p*-values is incomplete. Learning to recognize that interpretation of output critically depends on methods used & the pipeline in which the results are obtained. Developing awareness of false discovery rate controls. Learning that the interpretation of their immediate results could be an interim step in an overall problem-solving context. | Seeks guidance to interpret results/ output, including considerations of alignment of methods & results. Understands that the *p*-value represents evidence about the null hypothesis, not the study hypothesis, but does not consistently avoid reification. Recognizes that, but does not always act as if, very small *p*-values are *not* "highly significant results". Can apply false discovery rate controls, but does so only when reminded/ required. Recognizes when the interpretation of their immediate results is an interim step in an overall problem-solving context. | Can discern, based on immediate results, methods & hypotheses, whether more experiments &/or data processing are required for robust result interpretation; collaboratively uses the appropriate knowledge base & data resources to interpret results; resists reification & is committed to good-faith efforts to falsify hypotheses. Consistently & appropriately uses false discovery rate controls. | Interprets own & others' results critically & with respect to the analysis plan; seeks/promotes alignment of methods, results, & interpretation. Prioritizes interpretable & reproducible results above any other outcome (e.g., publication or completion of tasks/project), & insists on false discovery rate controls & other sensitivity analyses in all work. Avoids problems that can arise in the interpretation of results, including bias, reification, & other failures of positivism. Is able to evaluate the quality & appropriateness of procedures, statistical analyses, & models when reviewing papers & projects/ proposals, based on the writers'–& own—interpretation of results. |

(*Continued*)

**Table 1.** (Continued)

| Performance Level: | Novice | Beginner | Apprentice | J1 Journeyman | J2 Journeyman |
|---|---|---|---|---|---|
| **Draw & contextualize conclusions** | Does not draw appropriate conclusions from given results; without direction, will not even contextualize conclusions with the protocol that was followed. Not aware of the difference between conclusions about the null hypothesis & those about the research hypothesis. Conclusions may over- or under-state results & be driven by *p*-values or other superficial cues. Does not recognize the importance of identifying & acknowledging methodological limitations, or their implications, for conclusions. Does not or cannot apply rules of logic to scientific arguments, & commits logical fallacies when drawing conclusions. | Learning fundamentals of how appropriate conclusions are drawn from results, but may not be able to draw those conclusions from given results themselves. Learning to differentiate between conclusions about the null hypothesis & those about the research hypothesis. Learning why *p*-value-driven conclusions, & the lack of false discovery rate controls, are not conducive to reproducible work. Conclusions are generally aligned with given results, but when multiple methods are used, does not recognize the dependencies among methods that appear to reinforce, but actually replicate, results. Conclusions are neither fully contextualized with the rest of a document (write-up, paper, etc) or study/ experiments/paradigm (contextualization for *coherence*), nor with the literature (*critical contextualization*). | With guidance, can draw conclusions in own work that are coherent with the research hypothesis/hypotheses & across the entire manuscript/writeup (as appropriate). Learning to critically contextualize results; is able to draw the most obvious conclusions, but struggles to see patterns, or draw more subtle conclusions. Learning that "full"contextualization of conclusions requires consideration of limitations deriving from methods & their applications, & their effects on results & conclusions. Learning to recognize how independence of multiple methods applied to similar data/problems supports reproducible conclusions. | Can extract scientific meaning from data analysis & knows the difference between statistical & biological significance. In their own & others' work, seeks competing, plausible alternative conclusions. Can judge the scientific importance of their results, & draws conclusions accordingly. Can draw conclusions & contextualize results with respect to an entire manuscript/writeup in a given project or study, or with literature (as appropriate). Can detect when conclusions are not aligned with other aspects of the work (e.g., introduction/ background, methods &/or results, or other experiments in the project). Gives careful consideration to limitations deriving from the method & its application in a specific study. Sees patterns, & perceives more subtle conclusions than earlier-stage scientists, & collaborates to fully articulate & motivate them. Writes the Discussion & Conclusions sections, including limitations, of own articles, with collaboration. | Expertly contextualizes results & conclusions with prior literature, across experiments or studies, & within any given document (e.g., manuscript, writeup, etc.). Strives to fully contextualize conclusions in own work, & also requires this in others' work. Draws & contextualizes more subtle conclusions than at earlier stages. Can conceptualize new experiments based on the lack of robust &/or defensible conclusions in others' work. Carefully considers consistency of conclusions with the other parts of own or others' work. |
| **Communication** | Does not communicate scientific information clearly or consistently; is unaware of community standards for scientific communication. Generally relies on lay summaries to support own communication; does not recognize that using original literature strengthens scientific communication. Does not differentiate appropriate & inappropriate scientific communication, nor understand the ethical implications of each. | Learning both to recognize the value of clear communication, & about the role of communication in sharing & publishing research, data, code, data management, tools & resources. Developing an awareness of community standards for scientific communication, & that these include documenting code, annotating data, & adding appropriate metadata. Does not adapt communication to fit the receiver. Learning to differentiate appropriate & inappropriate scientific communication, but does not yet understand that transparency in all communication represents ethical practice, *even when* the desired results have not been achieved. | Understands the roles of sharing & publishing research, data, code, data management, tools & resources in scientific communication. Seeks guidance so that own communication is coherent, accurate, & consistent with community standards (e.g., following FAIR‡ principles; ensuring socially responsible science). Learning to document code, annotate data, & add appropriate metadata–& the importance of these (as appropriate given their research/context) for sharing & integration. Learning the importance of adapting communication to fit the receiver, seeking opportunities to practice this. Learning that transparency in all communication represents ethical practice, even when the desired results have not been achieved. | Consistently & proficiently uses technical language to correctly describe what was done, why, & how. Sufficient consideration given to limitations, with explicit contextualization of results consistently included in the communication of results & their interpretation. Can adapt communication to fit the receiver; recognizes that sometimes communication must be consistent with community standards beyond their own discipline. Appropriately documents/annotates all data, code, tools, & resources for sharing, integration, & re-use. Understands that transparency in all communication represents ethical practice. | Is an expert communicator & reviewer of scientific communication; adheres to & promotes disciplinary standards for communication. Communicates in a manner that is consistent with standards across communities beyond their own discipline, as appropriate. Ensures communication is appropriate for a target audience, expertly adapting to fit the receiver(s). Communication is transparent, & appropriate to support reproducibility–& thereby, ethical practice—in every context. |

*Framework of the workflow supports decisions; workflow is not necessarily linear and can be multidirectional and iterative; any point can be re-iterated, or new starts from within the workflow can be made. A pipeline is unidirectional, not iterative within its structure (it is ballistic: once initiated, it runs), and has no decision points. Pipelines can exist within workflows, but workflows do not exist in pipelines.

‡ FAIR: Findable, Accessible, Interoperable, and Reusable.

## Validity evidence

The first step in validating the MR-Bi was to investigate the alignment of the KSAs with the competencies. While the competencies informed the refinement of the KSAs (see Supplemental Materials, Figure A in S1 File), our emphasis was nevertheless on the scientific method. To support the claim that the MR-Bi is valid for the domain, the final KSAs should support the competencies such that, if a curriculum develops the KSAs in learners to a sufficient level, then performance of the relevant KSA(s) should lead to the demonstration of target competencies. This alignment is presented in Table 2, and assumes that each competency would be assessed as "present"/"absent": i.e., that all of its constituent parts are required for a person to be declared "competent" for that item. To perform the alignment, we took the highest Bloom's

**Table 2. Alignment of KSAs with competencies for bioinformatics and biomedical informatics.**

| COMPETENCIES: where Bloom's cognitive taxonomy is clearly invoked, these are identified in italics | KSAs | | | | | | | | | | |
|---|---|---|---|---|---|---|---|---|---|---|---|
| | Prerequisite Knowledge-bio | Prerequisite Knowledge-comp | Interdisc Integr | Define a problem | Hypothesis generation | Experi-mental Design | Identify data | Identify and use methods | Interpret results/ output | Draw and context-ualize con-clusions | Commun-icate |
| Bioinformatics (Welch et al., 2014) | | | | | | | | | | | |
| 1. Ability to *apply* knowledge of computing, biology, statistics & mathematics appropriate to the discipline | x | x | x | | | | | x | | | |
| 2. *Knowledge* of general biology, in-depth knowledge of at least one area of biology, & *understanding* of biological data-generation technologies | x | | | | | | | | | | |
| 3. Ability to *analyze* a problem, & *identify & define* the computing requirements appropriate to its solution | ‡ | x | ‡ | ‡ | | | | ‡ | | | |
| 4. Ability to *apply* mathematical foundations, algorithmic principles & computer science theory to the modeling & design of computer-based systems in a way that demonstrates comprehension of the trade-offs involved in design choices | | x | x | | | x | | x | | | |
| 5. Ability to *design, implement & evaluate* a computer-based system, process, component or program to meet desired needs in scientific environments | | x | x | | | | | x | | | x |
| 6. Ability to apply design & development principles in the *construction* of software systems of varying complexity | | x | | | | | | x | | | |
| 7. Ability to *use* current techniques, skills & tools necessary for computational biology practice | x | x | x | x | x | x | x | x | x | x | x |
| 8. Ability to function effectively on teams to accomplish a common goal | | | x | | | | | | | | x |

(*Continued*)

**Table 2.** (Continued)

| COMPETENCIES: where Bloom's cognitive taxonomy is clearly invoked, these are identified in italics | KSAs | | | | | | | | | | |
|---|---|---|---|---|---|---|---|---|---|---|---|
| | Prerequisite Knowledge-bio | Prerequisite Knowledge-comp | Interdisc Integr | Define a problem | Hypothesis generation | Experimental Design | Identify data | Identify and use methods | Interpret results/output | Draw and context-ualize con-clusions | Commun-icate |
| 9. *Understanding* of professional, ethical, legal, security & social issues & responsibilities ** | | | | | * § | | | | | | |
| 10. Ability to communicate effectively with a range of audiences | | | x | | | | | | | | x |
| 11. Ability to *analyze* the local & global impact of bioinformatics & genomics on individuals, organizations & society | x§ | x§ | x§ | x§ | x§ | x§ | x§ | x§ | x§ | x§ | x |
| 12. Recognition of the need for, & ability to engage in, CPD | | | | | * | | | | | | |
| 13. Detailed *understanding* of the scientific discovery process & of the role of bioinformatics in it | x | x | x | x | x | x | x | x | x | x | x |
| 14. Ability to *apply* statistical research methods in the contexts of molecular biology, genomics, medical & population genetics research | | | | | ∞ | | | | | | |
| **Core competencies for <u>doctoral</u> health/medical informatics (Kulikowski et al. 2012; emphasis added)** | | | | | | | | | | | |
| **Fundamental scientific skills** | | | | | | | | | | | |
| Acquire professional perspective: *understand & analyze* the history & values of the discipline, & its relationship to other fields, while demonstrating an ability to read, *interpret & critique* the core literature | x | x | x | | | | | | x | x | x |
| Analyze problems: *analyze, understand, abstract & model* a specific biomedical problem in terms of data, information & knowledge components | x | x | x | | | | x | x | | | x |
| Produce solutions: use the problem analysis to identify & understand the space of possible solutions & *generate designs* that capture essential aspects of solutions & their components | | x | x | | | | | | | x | x |

(*Continued*)

**Table 2.** (Continued)

| COMPETENCIES: where Bloom's cognitive taxonomy is clearly invoked, these are identified in italics | KSAs | | | | | | | | | | |
|---|---|---|---|---|---|---|---|---|---|---|---|
| | Prerequisite Knowledge-bio | Prerequisite Knowledge -comp | Interdisc Integr | Define a problem | Hypothesis generation | Experi-mental Design | Identify data | Identify and use methods | Interpret results/ output | Draw and context-ualize con-clusions | Commun-icate |
| Articulate the rationale: *defend* the specific solution & its advantage over competing options | x | x | x | | | | | x | | x | x |
| *Implement, evaluate & refine*: carry out the solution (including obtaining necessary resources & managing projects), evaluate it, & iteratively improve it | x | x | x | | x | x | x | x | x | x | x |
| *Innovate*: *create* new theories, typologies, frameworks, representa-tions, methods & processes to address biomedical informatics problems | x | x | x | x | x | x | x | x | x | x | x |
| Work collaboratively: team effectively with partners within & across disciplines | | | x | | | | | | | | x |
| Educate, disseminate & *discuss*: communicate effectively to students & other audiences in multiple disciplines in persuasive written & oral form | | | x | | | | | | | | x |
| **Scope and breadth of the discipline** | | | | | | | | | | | |
| Prerequisite knowledge & skills: students must *be familiar* with biological, biomedical & population-health concepts & problems, including common research problems | x | | | | | | | | | | |
| Fundamental knowledge: *understand the fundamentals* of the field in the context of the effective use of biomedical data, information & knowledge | x | x | x | | | | | | | | |
| – Biology: molecule, sequence, protein, structure, function, cell, tissue, organ, organism, phenotype, populations. | x | x | x | | | | | | | | |

(*Continued*)

**Table 2.** (Continued)

| COMPETENCIES: where Bloom's cognitive taxonomy is clearly invoked, these are identified in italics | KSAs | | | | | | | | | | |
|---|---|---|---|---|---|---|---|---|---|---|---|
| | Prerequisite Knowledge-bio | Prerequisite Knowledge-comp | Interdisc Integr | Define a problem | Hypothesis generation | Experi-mental Design | Identify data | Identify and use methods | Interpret results/ output | Draw and context-ualize con-clusions | Commun-icate |
| – Translational and clinical research: genotype, phenotype, pathways, mechanisms, sample, protocol, study, subject, evidence, evaluation. | x | x | x | | | | | | x | x | |
| – Healthcare: screening, diagnosis (diagnoses, test results), prognosis, treatment (medications, procedures), prevention, billing, healthcare teams, quality assurance, safety, error reduction, comparative effectiveness, medical records, personalized medicine, health economics, information security and privacy. | x | x | x | | | | x | | x | x | |
| – Personal health: patient, consumer, provider, families, health promotion, personal health records. | x | x | x | | | | | | | | |
| – Population health: detection, prevention, screening, education, stratification, spatio-temporal patterns, ecologies of health, wellness. | x | x | x | | | | | | | | |
| **Procedural knowledge and skills** | | | | | | | | | | | |
| For substantive problems related to scientific inquiry, problem solving & decision-making, *apply, analyze, evaluate & create* solutions based on biomedical informatics approaches | x | x | x | | x | x | x | x | x | x | x |
| *Understand & analyze* complex biomedical informatics problems in terms of data, information & knowledge | x | x | x | | | | x | x | | | x |
| *Apply, analyze, evaluate & create* biomedical informatics methods that solve substantive problems within & across biomedical domains | x | x | x | x | x | x | x | x | x | x | x |

(*Continued*)

**Table 2.** (Continued)

| COMPETENCIES: where Bloom's cognitive taxonomy is clearly invoked, these are identified in italics | KSAs | | | | | | | | | | |
|---|---|---|---|---|---|---|---|---|---|---|---|
| | Prerequisite Knowledge-bio | Prerequisite Knowledge-comp | Interdisc Integr | Define a problem | Hypothesis generation | Experi-mental Design | Identify data | Identify and use methods | Interpret results/output | Draw and context-ualize con-clusions | Commun-icate |
| *Relate* such knowledge & methods to other problems within & across levels of the biomedical spectrum | x | x | x | x | | | x | x | | x | x |
| **Theory and methodology** | | | | | | | | | | | |
| Theories: *understand & apply syntactic*, semantic, cognitive, social & pragmatic theories as they are used in biomedical informatics | | | x | | | | | x | | | |
| Typology: *understand & analyze* the types & nature of biomedical data, information & knowledge | x | x | x | | | | x | x | | | |
| Frameworks: *understand & apply* the common conceptual frameworks used in biomedical informatics (e.g., including belief networks, programming approach, representational scheme or architectural design) | | x | x | | | | | | | | |
| Knowledge representation: *understand & apply* representations & models that are applicable to biomedical data, information & knowledge (knowledge representation is a method of encoding concepts & relationships within a domain using definitions that are computable | x | x | x | | | | | x | | | |
| Methods & processes: *understand & apply existing* methods & processes used in different contexts of biomedical informatics | | x | | | | | | | | | |
| **Technological approach** | | | | | | | | | | | |
| Prerequisite knowledge & skills: assumes *familiarity* with data structures, algorithms, programming, mathematics, statistics | | x | | | | | | | | | |

*(Continued)*

**Table 2.** (Continued)

| COMPETENCIES: where Bloom's cognitive taxonomy is clearly invoked, these are identified in italics | KSAs | | | | | | | | | | |
|---|---|---|---|---|---|---|---|---|---|---|---|
| | Prerequisite Knowledge-bio | Prerequisite Knowledge-comp | Interdisc Integr | Define a problem | Hypothesis generation | Experi-mental Design | Identify data | Identify and use methods | Interpret results/ output | Draw and context-ualize con-clusions | Commun-icate |
| Fundamental knowledge: *understand & apply* technological approaches in the context of biomedical problems: For example: | x | x | x | | | | | x | x | x | x |
| – Imaging and signal analysis. | x | x | x | | | | x | x | | x | x |
| – Information documentation, storage, and retrieval. | x | x | x | | | | x | x | | x | x |
| – Machine learning, including data mining. | x | x | x | | | x | x | x | x | x | x |
| – Networking, security, databases. | x | x | x | | | | x | x | x | x | x |
| – Natural language processing, semantic technologies. | x | x | x | | | | x | x | x | x | x |
| – Representation of logical and probabilistic knowledge and reasoning. | x | x | x | | | | x | x | x | x | x |
| – Simulation and modeling. | x | x | x | | x | x | x | x | x | x | x |
| – Software engineering. | x | x | x | | x | x | x | x | x | x | x |
| Procedural knowledge & skills: for substantive problems, *understand & apply* methods of inquiry & criteria for selecting & utilizing algorithms, techniques & methods | x | x | x | | | x | | x | | x | x |
| *Describe* what is known about the application of the fundamentals within biomedicine | x | x | | | | | | | | | x |
| *Identify the relevant existing approaches for a specific biomedical problem* | *x* | *x* | *x* | | | | | *x* | | | |
| *Apply, adapt & validate an existing approach to a specific biomedical problem* | | *x* | | | *x* | *x* | | *x* | *x* | *x* | *x* |
| Human and social context *§ | | | | | | | | | | | |
| **Human and social context**: draws upon the social and behavioral sciences to inform the design and evaluation of technical solutions, policies, and the evolution of economic, ethical, social, educational, and organizational systems. | | x* | x* | | | | | | | | |

*(Continued)*

**Table 2.** (Continued)

| COMPETENCIES: where Bloom's cognitive taxonomy is clearly invoked, these are identified in italics | KSAs | | | | | | | | | | |
|---|---|---|---|---|---|---|---|---|---|---|---|
| | Prerequisite Knowledge-bio | Prerequisite Knowledge -comp | Interdisc Integr | Define a problem | Hypothesis generation | Experi-mental Design | Identify data | Identify and use methods | Interpret results/ output | Draw and context-ualize con-clusions | Commun-icate |
| Prerequisite knowledge and skills: Familiarity with fundamentals of social, organizational, cognitive, and decision sciences. | | | x | | | | | | | | |
| **Fundamental knowledge: Understand and apply knowledge in the following areas**: | | | | | | | | | | | |
| – Social, behavioral, communication, and organizational sciences: for example, computer supported cooperative work, social networks, change management, human factors engineering, cognitive task analysis, project management. | | x | x | | | | | | | | |
| – Ethical, legal, social issues: for example, human subjects, HIPAA, informed consent, secondary use of data, confidentiality, privacy. | | x | x | | | | | | | | |
| – Economic, social and organizational context of biomedical research, pharmaceutical and biotechnology industries, medical instrumentation, healthcare, and public health. | | x* | x* | | | | | | | | |
| **Procedural knowledge and skills (human and social context)** | | | | | | | | | | | |
| – *Apply, analyze, evaluate, and create* systems approaches to the solution of substantive problems in biomedical informatics. | x | x | x | x | x | x | x | x | x | x | x |
| – Analyze complex biomedical informatics problems in terms of people, organizations, and socio-technical systems. | | | x | | | | | | | | x |
| – *Understand* the challenges and limitations of technological solutions. | x | x | x | | | | | | x | | |

(*Continued*)

**Table 2.** (Continued)

| COMPETENCIES: where Bloom's cognitive taxonomy is clearly invoked, these are identified in italics | KSAs | | | | | | | | | | |
|---|---|---|---|---|---|---|---|---|---|---|---|
| | Prerequisite Knowledge-bio | Prerequisite Knowledge -comp | Interdisc Integr | Define a problem | Hypothesis generation | Experi-mental Design | Identify data | Identify and use methods | Interpret results/ output | Draw and context-ualize con-clusions | Commun-icate |
| – *Design and implement* systems approaches to biomedical informatics applications and interventions. | x | x | x | | | x | | | | x | |
| – *Evaluate* the impact of biomedical informatics applications and interventions in terms of people, organizations, and socio- technical systems. | | | x | | | | | | x | x | x |
| – *Relate* solutions to other problems within and across levels of the biomedical spectrum. | x | x | x | x | x | | | | | x | x |

x = the indicated KSA must be applied at some level (of Bloom's taxonomy) in order to accomplish the competency.

‡ It is unclear whether "computing requirements" in Bioinformatics competency # 3 refers to simple computer hardware and software, or also includes methods, techniques and other–more challenging to identify & utilize–aspects of the requirements "appropriate to its solution". If this competency refers to an individual's minimal assessment of "appropriate", and if the problem for which computing is required has already been defined, then just one KSA (prerequisite knowledge, computing) is required. If the competency refers to an individual's evaluation of competing solutions, particularly if the problem for which the computing is required has not been defined a priori, then multiple KSAs are required. Neither the 2014 nor 2018 statements of the Bioinformaticscompetencies supports a distinction between these options.

§ Recognizing a need for a separate KSA in the MR-Bi that captures "ethical practice" was partly driven by Bioinformatics competency #9, but competency #9 is insufficiently specific to fully align KSAs. The possession of an understanding of ethical obligations, for example, do not translate to ethical practice, so even that KSA cannot be aligned with this competency. Similarly, Bioinformatics competency #11 suggests that "analysis" is sufficient to demonstrate the competency but this is incorrect. The ability to carry out this analysis also does not translate to ethical practice.

* no actionable verbs relating to the learner are included in the articulation of this competency or competency domain, so no KSAs could be justifiably aligned with this competency. More specifically, because there is no indication of what a learner needs to do to demonstrate this competency/competency domain (Bioinformatics competencies #9 & #12; medical informatics competency "human and social context" competencies), it cannot be taught or assessed in any systematic way. While the competencies are important, they are insufficiently specific to confidently align with any particular KSAs.

** This competency is insufficiently specific for determination of whether any KSAs are needed to achieve it. Because there is no indication of what a learner needs to do to demonstrate this competency/competency domain, it cannot be taught or assessed in any systematic way.

∞ This competency could be demonstrated at either the "apply" (low Bloom's level/few KSAs required), or at a considerably higher level. The level required to select, use, and interpret "statistical research methods", which as stated, is what doctoral level statisticians are trained to deploy (high Bloom's level/all KSAs required). The low-Bloom's level interpretation would not be contextualized in any of the specific research domains *by the performer*, while the higher level performer might actually specialize in just one of the contexts (e.g., biology or computation). Therefore, this competency is insufficiently specific for determination of which KSAs are needed to achieve it.

level of complexity described in the competency as the *minimum required* to successfully exhibit that competency.

The essential features of Table 2 for purposes of validating the MR-Bi are:

1. almost every competency is supported by at least one KSA; and

2. each KSA supports the achievement of at least one competency.

Table 2 suggests that the KSAs can support both sets of competencies, providing convergent as well as content validity. However, the table shows that several of the competencies were insufficiently articulated for alignment with any KSA. Two items (labeled with a * in Table 2) could not be aligned because they lacked actionable verbs: *Recognition of the need for and an ability to engage in continuing professional development* [19] and *Human and social context* [17]. Without actionable verbs, competencies cannot be taught or assessed in any systematic way. Three other items (labeled ** in Table 2) had potentially actionable verbs, but were insufficiently specified to align with any KSA: *An understanding of professional, ethical, legal, security and social issues and responsibilities* [19]; *Evaluate the impact of biomedical informatics applications and interventions in terms of people, organizations and socio-technical systems*; and *Analyze complex biomedical informatics problems in terms of people, organizations and socio-technical systems* [17]. In these cases, what constitutes the sufficient demonstration of the competencies is not discernable. By contrast, the PLDs in the MR-Bi contain actionable verbs describing the learner performing each KSA across stages, providing guidance to the learner about evidence that can demonstrate their achievements, and to the instructor about how to elicit such evidence.

## Alignment of the MR-Bi with the principles of andragogy

The second aspect of validation was to consider alignment of the MR-Bi with principles of andragogy [43]. This alignment is explored in Table 3. The results in Table 3 are derived from the design features of any Mastery Rubric but are also based on curriculum development, evaluation, or revision experiences in other Mastery Rubric development projects (i.e., [30]; [48]; [57], respectively).

The Mastery Rubric construct itself was created to facilitate the development of higher education curricula; the MR-Bi is thereby similarly aligned with these core principles.

## Discussion

Rubrics are familiar tools–often, scoring tables–to help instructors to evaluate the quality of, and hence to grade, individual pieces of student work in a consistent way. They typically contain quality descriptors for various evaluative criteria at specific levels of achievement [35]. When shared, so that instructors can use them for marking, and students can use them to plan/monitor/evaluate their own work, they have a positive impact on learning outcomes (e.g., [58]; [59]; [60]).

The Mastery Rubric builds on this concept, but with a focus on entire curricula or training programs rather than on discrete pieces of work. As such, a Mastery Rubric provides an *organising framework* in which KSAs are clearly articulated, and their performance levels described and staged in such a way that they can be achieved progressively. Specifically, the framework has three components: i) a set of domain-relevant, and transferable, KSAs; ii) a defined set of developmental stages denoting progression along a path of increasing cognitive complexity, towards independence (if that is desired); and iii) descriptions of the range of expected performance levels of those KSAs. The interplay between these 'dimensions' affords the Mastery Rubric significant flexibility. In particular, it recognizes that individuals may be at different levels in different KSAs, and hence may have different speeds of traversal through them—thus, importantly, the measure of progression is not time, as in traditional educational systems [61] but rather, demonstrable acquisition of specific KSAs. This allows individuals (students, technicians or PIs) who wish to acquire bioinformatics skills to locate themselves within the matrix regardless of their current skill level or disciplinary background: for example, a person may be an Early Journeyman (J1) in biology (i.e., has earned a doctorate), yet be a Novice in

**Table 3. How the principles of andragogy would be met with a Mastery Rubric like the MR-Bi.** Table adapted from [57], Table 4, with permission.

| | Principle of andragogy | Met with a Mastery Rubric? |
|---|---|---|
| 1 | *Adults are self-directed and internally motivated, and so can—and need to—take responsibility for choices that further learning objectives.* | The Mastery Rubric is *shared*. The KSAs direct the learner to relevant learning targets, while the PLDs show the levels at which their achievement should be demonstrated. The developmental trajectory enables learners to make strategic decisions throughout a curriculum, within academic courses, or in training programs, facilitating their ongoing growth. |
| 2 | *Adults bring prior knowledge and experience to learning–and seek to connect new information with prior learning.* | The Mastery Rubric allows learners to recognize and demonstrate where they are (how they perform) in their development, and to motivate their acquisition of, and organize, new knowledge. This enables them to leverage their prior knowledge and experience, supporting self-direction along the articulated trajectory. |
| 3 | *Adults are goal oriented, and require explicit, recognizable, and achievable learning goals.* | The goals of learning are expressed in all three dimensions (KSAs, developmental stages, PLDs) of a Mastery Rubric. PLDs present concrete and recognizable targets that learners can see how to achieve. Courses and series of courses can be selected by learners (and aligned with PLDs by instructors) to ensure accomplishment of curriculum or course goals. |
| 4 | *Adult learners need to know <u>why</u> they are learning what is presented.* | The KSAs make explicit what learners need to learn to support both current and future practice. The trajectory offers justification for ongoing learning, while the PLDs describe readiness for the next learning goal. Together, these provide learners with a rationale for commitment and engagement throughout a curriculum. |
| 5 | *Adult learners benefit from practical, authentic assessments and practice experience.* | The Mastery Rubric supports instruction and curriculum design that emphasize authenticity and transferability of new knowledge (e.g., to new contexts) through: 1) KSAs that are relevant; 2) trajectories that support evolution through recognizable and practical stages; and 3) PLDs that represent the development of observable behaviors. Together, these three dimensions facilitate learners' selection of curricula, programs, or courses that support both current and future practice in the domain. |
| 6 | *Adults are motivated to learn but need to be treated as partners in the learning enterprise, not as vessels to be filled.* | The Mastery Rubric is intended to function as a *contract* between instructor and learner. Its three dimensions are intended to provide sufficient context to define and justify the responsibilities of learners *and* instructors as full partners in the learning enterprise. |

bioinformatics; or a person may be an Apprentice-level computer scientist or 'engineer' yet a bioinformatics Beginner. For all learners, the MR-Bi not only identifies where they are, but it also makes the route from their current level of performance on any given KSA to a higher level *explicit*, without the need to articulate bespoke personal traits for individuals from every conceivable scientific background.

A strength of the MR-Bi is that it was developed and refined specifically to support decisions that are made by instructors and learners, and to promote and optimize educational outcomes. In this way, the MR-Bi can bring *validity*—a formal construct relating to the decisions that are supported by any test, score, or performance [62]; [63]—to bioinformatics curriculum or training program development. Although it does not focus on subject-specific content, the MR-Bi should lead to curricula that produce similarly-performing graduates (or course completers) across institutions; that is, a curriculum or training program that uses the MR-Bi can

go beyond a transcript of what courses a learner completed, to represent what a learner can do and the level at which they perform. This is consistent with the European Qualifications Framework (see https://ec.europa.eu/ploteus/content/how-does-eqf-work), which defines eight hierarchical levels describing what learners know, understand, and are able to do, aiming to make it easier to compare national qualifications and render learning transferable (https://ec.europa.eu/ploteus/content/descriptors-page). The European Qualifications Framework enables mapping of the disparate characterizations of high school graduates, university graduates, and doctorate awardees into a single, coherent set of general descriptors. The MR-Bi makes this type of mapping specifically developmental.

The orientation of the MR-Bi towards a particular definition of bioinformatics education [55] and, by extension, bioinformatics practice, should be recognized. The context in which MR-Bi users *principally* focus is assumed to be the life sciences; more specifically, that their ultimate learning goals are oriented towards *solving biological problems* using computational technology/techniques. Moreover, the developmental trajectories outlined by the PLDs, and the choice of focal KSAs, support bioinformatics education and training that seek to move individuals towards *independence in their practice of–or contributions to—science*. Accordingly, the KSA-extraction process was heavily influenced by models of the scientific method and scientific reasoning (following [44] and [45]). It is therefore important to emphasize that using the methods described here, other investigators might generate different KSAs *if neither independence nor the scientific method are essential to their objectives*.

Further, although we considered competencies for both bioinformatics and medical/health informatics, the PLDs for the MR-Bi were devised by bioinformaticians–for bioinformaticians. If the goal were to derive a Mastery Rubric specific for medical/health informatics, and scientific independence was similarly important, then the same process of PLD development described here could be used; the resulting Mastery Rubric would have virtually the same KSAs, but with *Prerequisite knowledge—biology*, replaced by *Prerequisite knowledge–medical informatics/health informatics/health systems* (as appropriate). The PLDs in that Mastery Rubric would then be tailored to describe achievement and development of the health informatics practitioner.

It is also worth noting that the high-level stage descriptions (top of Table 1), which informed the PLD-drafting process, broadly track the *typical* development of an undergraduate who progresses to graduate school. It could be argued that these definitions are too rigid. After all, individuals differ in their motivation, and in their capacities and *attitudes* towards learning and growth; thus, in the 'real world', an undergraduate class may include students with behaviours and attitudes characteristic of Novice and Beginner levels of cognitive complexity (or, exceptionally, Apprentice level). Hence, if challenged with advanced training methods (e.g., introducing novel research projects into undergraduate biology curricula), some students in the class will be receptive to being pushed beyond their intellectual 'comfort zones', while many will resist, being more comfortable with lectures and "canned" exercises with known results (e.g., [64]). Nevertheless, the broad descriptions in Table 1 were ultimately those that resonated with our own practical experiences of bioinformatics education and training at all stages, and for pragmatic purposes the mapping was therefore necessarily general, like any true rubric.

It should be stressed that creating any kind of 'framework' to support the development of competencies involves numerous stakeholders and is hugely time-consuming, not least because marrying multiple stakeholder views is hard. For example, the current version of the European e-Competence Framework for Information and Computing Technology skills (http://www.ecompetences.eu) has taken more than 8 years to develop; similarly, the bioinformatics competencies have been evolving over at least the last 6 years–inevitably, few aspects

escape dissent when stakeholders from very different backgrounds, with disparate student populations and different educational goals attempt to achieve consensus [65]. Likewise, formulation of the current version of the MR-Bi has taken more than 2 years, and further refinements are likely to be required with future additional stakeholder input.

Amongst the many challenges for initiatives developing competency frameworks is the lack of a standard vocabulary. In consequence, while the MR-Bi describes knowledge, skills and *abilities*, the European e-Competence Framework refers to knowledge, skills and *attitudes*, and the BioExcel competency framework to knowledge, skills and *behaviours* [66]. We use *abilities* because they are observable and connote more purposeful engagement than do attitudes or behaviours, and this is the terminology used in discussions of validity in education/educational assessment (e.g., see [53]).

Mastery Rubrics have strengths that can be leveraged by institutions, instructors, and scientists. They support curriculum development in any education program, making concrete and explicit the roles–and contributions–of learner and instructor. Since it was developed as a tool for *curriculum* development, it may be difficult to conceptualize how the MR-Bi can be used to support the development of short training programs or courses. Nevertheless, even in the absence of a formal curriculum, for stand-alone and/or linked courses, the Mastery Rubric can help i) instructors to focus on prerequisite knowledge and learning objectives that are time-delimited; and, ii) learners to identify targeted training opportunities and thence to track their personal/professional development with the PLDs and KSAs.

For example, a possible application of the MR-Bi could be the classification (or development) of teaching/training materials and courses according to which KSAs they support, where learners should start (at which stage of each KSA) in order to benefit optimally from the training, and to which stage a given training opportunity proposes to bring them. Another important application of this tool could be the revision of existing courses and materials so that they explicitly support learning/demonstration of targeted KSAs at a given level. The integration of MR-Bi features into teaching materials would both provide guidance to the self-directed learner, and support the standardization of teaching and learning goals across instructional materials, and across formal and informal bioinformatics training programs worldwide. The MR-Bi is therefore a timely contribution to current global conversations and initiatives (including ELIXIR's Training e-Support System [67], [68], which is championing the uptake of Bioschemas.org specifications for sharing training materials [69]; the training portal of the Global Organization for Learning, Education, and Training [70]; the Educational Resource Discovery Index [71]; and The Carpentries [72]) about standardizing and sharing training resources, and best practices for personalizing and customizing learning experiences.

## Conclusions

In recent years, concern about the growing computational skills gap amongst life scientists has prompted the articulation of core bioinformatics competencies, aiming to facilitate development of curricula able to deliver appropriate skills to learners. However, implementing competencies in curricula has proved problematic: this is partly because there are still disparate views on what it means to be a trained bioinformatician, and partly also because competencies are actually complex, multi-dimensional educational end-points, making it difficult to achieve a common understanding of how to deliver requisite training in practice (e.g., [65]; [56]). These problems have led some researchers to suggest refinements to bioinformatics competencies, including the identification of phases of competency, and the provision of guidance on the evidence required to assess whether a given competency has been acquired [20]; others have

already revised their competency-based framework to include learning trajectories, charting the progression of individuals' abilities through defined stages via milestones [28].

Cognisant of these issues, we have devised a Mastery Rubric—a formal framework that supports the development of specific KSAs, and provides a structured trajectory for achieving bioinformatics competencies. Importantly, it prioritizes the development of independent scientific reasoning and practice; it can therefore contribute to the cultivation of a next generation of bioinformaticians who are able to design rigorous, reproducible research, and critically analyze their and others' work. The framework is inherently robust to new research or technology, because it is broadly content agnostic. It can be used to strengthen teaching and learning, and to guide both curriculum building and evaluation, and self-directed learning; any scientist, irrespective of prior experience or disciplinary background, can therefore use it to document their accomplishments and plan further professional development. Moreover, the MR-Bi can be used to support short training courses by helping instructors to focus on prerequisite knowledge and on learning objectives that are time-delimited. Specifics on how to accomplish these are the topics of our ongoing work.

## Supporting information

**S1 File. Cognitive Task Analysis Methodology (Table A) and Figure (Figure A) with examples (Text A), and full listing of all competencies (Table B).**
(DOCX)

## Acknowledgments

The authors gratefully acknowledge the important contributions of the National Bioinformatics Infrastructure for Sweden (www.nbis.se) informatics instructors who attended the standard-setting meeting at the national center for molecular biosciences, Science for Life Laboratory (SciLifeLab, www.scilifelab.se) and the Swedish National Bioinformatics platform in Stockholm. We gratefully acknowledge the Georgetown University Department of Neurology, the GOBLET Foundation, and ELIXIR Italy for defraying the publication costs of this article.

## Author Contributions

**Conceptualization:** Rochelle E. Tractenberg.

**Data curation:** Rochelle E. Tractenberg, Jessica M. Lindvall, Teresa K. Attwood, Allegra Via.

**Formal analysis:** Rochelle E. Tractenberg, Jessica M. Lindvall, Teresa K. Attwood, Allegra Via.

**Investigation:** Rochelle E. Tractenberg.

**Methodology:** Rochelle E. Tractenberg.

**Project administration:** Rochelle E. Tractenberg.

**Validation:** Rochelle E. Tractenberg.

**Visualization:** Rochelle E. Tractenberg.

**Writing – original draft:** Rochelle E. Tractenberg, Jessica M. Lindvall, Teresa K. Attwood, Allegra Via.

**Writing – review & editing:** Rochelle E. Tractenberg, Jessica M. Lindvall, Teresa K. Attwood, Allegra Via.

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
