## [Decision Letter · Decision Letter 0]

3 Sep 2019

PONE-D-19-15839

The Mastery Rubric for Bioinformatics: supporting design and evaluation of career-spanning education and training

PLOS ONE

Dear Professor Tractenberg,

Thank you for submitting your manuscript to PLOS ONE. After careful consideration, we feel that it has merit but does not fully meet PLOS ONE’s publication criteria as it currently stands. Therefore, we invite you to submit a revised version of the manuscript that addresses the points raised during the review process.

I agree with both reviewers in their assessment that the manuscript is somewhat dense and that a lot of acroynms are used. Perhaps it would make for an easier read if "KSA" were replaced with compentencies throughout and when KSA is mentioned on page 17 (with [22] cited) you could say something like "(KSAs, hereafter referred to as competencies)" or similarly for PLDs on Page 20 "(PLDs; [36], hereafter referred to as descriptors)". Likewise, in Table 2, I think you can spell out prerequisite knowledge instead of having PK in the table header. These are just a few examples for improvement. I would also recommend a figure as Reviewer 2 suggests, for helping readers to understand how your MR-Bi was arrived at.

We would appreciate receiving your revised manuscript by Oct 17 2019 11:59PM. To enhance the reproducibility of your results, we recommend that if applicable you deposit your laboratory protocols in protocols.io, where a protocol can be assigned its own identifier (DOI) such that it can be cited independently in the future. For instructions see: http://journals.plos.org/plosone/s/submission-guidelines#loc-laboratory-protocols

We look forward to receiving your revised manuscript.

Kind regards,

Nicholas J. Provart, Ph.D., M.Sc.

Academic Editor

PLOS ONE

Journal Requirements:

1. Thank you for including your competing interests statement; "Rochelle Tractenberg has read the journal's policy and has the following competing interest: She is a section editor for PLOS ONE. The other authors have declared that no competing interests exist."

Reviewers' comments:

Reviewer's Responses to Questions

**Comments to the Author**

1. Is the manuscript technically sound, and do the data support the conclusions?

Reviewer #1: Yes

Reviewer #2: Yes

2. Has the statistical analysis been performed appropriately and rigorously? 

Reviewer #1: N/A

Reviewer #2: N/A

3. Have the authors made all data underlying the findings in their manuscript fully available?

Reviewer #1: Yes

Reviewer #2: Yes

4. Is the manuscript presented in an intelligible fashion and written in standard English?

Reviewer #1: No

Reviewer #2: Yes

5. Review Comments to the Author

Reviewer #1: The authors describe the process of creating and validating a mastery rubric for bioinformatics training programs. The validation was thoroughly checked; they assessed whether it aligned with previously identified bioinformatics competencies as well as the principles of andragogy. The authors explain the problems that have been encountered while trying to create bioinformatics curricula using currently available bioinformatics competencies and discuss how this mastery rubric will solve some of these challenges. This paper is useful for anyone creating a training program in bionformatics. It's scope also extends beyond that, and is a concrete example of how mastery rubrics can be created and of general use for any program/discipline.

To increase the readability (which will affect the impact of this paper) there are two main concerns that should be addressed:

1. The authors use far far too many acronyms. I understand acronyms reduce manuscript length, however, they are difficult for the reader to remember and thus negatively influence readability of the manuscript. The number of acronyms in this paper resulted in a much greater reading time for this manuscript compared to other manuscripts of similar length that I have read in the past. I also had to resort to drafting a acronym cheatsheet for myself so I could read this coherently.

2. In many instances in this manuscript, sentences are overly long and complex.

Reviewer #2: In this manuscript entitled “The Mastery Rubric for Bioinformatics: supporting design and evaluation of career- spanning education and training”, Tractenberg and colleagues have submitted and summarized an important quantity of work dealing with the aspect of developing and describing a metric for the development of a bioinformatics education (undergraduate and graduate) and training (later specialized workshops) programs. The submission is well written, in a slightly dense format, with well written english.

The methodology used involved the development of a Mastery Rubric for Bioinformatics, created to address the challenges of integrating competencies into actionable teaching and learning goals. To do this they defined “Knowledge, Skills and Abilities” (KSAs) that are generic enough to be used in a number of scientific fields, but are specific here to the challenge of measuring and evaluating bioinformatics skills. These can then be used ro generate curricula for various levels of bioinformatics training.

Concerns:

1. The way the data was gathered, used and analyzed is described in some details, but it would be very useful to also add a figure presenting the workflow of this process. Also maybe abstracting the figure from one of the annexes.

2. Although this was by their own words not a quantitative analysis, some numbers on how many shared KSAs that characterize the scientific method were used, that were later shown to be “essential to bioinformatics education and training”.

3. This is presented as a “tool”, but it is clear no tool is presented. How an academic needs to work through this decision tree to see how her or his course is aligned to the KSAs and the various career stages is not clearly laid-out. A bit more of a a description how this could be best performed needs more details.

4. Adoption of this method in real life should have also been described, if it exists.

6. PLOS authors have the option to publish the peer review history of their article (what does this mean?). If published, this will include your full peer review and any attached files.

Reviewer #1: No

Reviewer #2: No

---

## [Editor Report · Decision Letter 1]

1 Nov 2019

The Mastery Rubric for Bioinformatics: a tool to support design and evaluation of career-spanning education and training

PONE-D-19-15839R1

Dear Dr. Tractenberg,

Thank you for your revisions - the manuscript reads much better with fewer acronyms and the supplemental figure helps with understanding how MR-Bi was developed. We are pleased to inform you that your manuscript has been judged scientifically suitable for publication and will be formally accepted for publication once it complies with all outstanding technical requirements.

With kind regards,

Nicholas J. Provart, Ph.D., M.Sc.

Academic Editor

PLOS ONE
---

## [Editor Report · Acceptance letter]

14 Nov 2019

PONE-D-19-15839R1 

The Mastery Rubric for Bioinformatics: a tool to support design and evaluation of career-spanning education and training 

Dear Dr. Tractenberg:

I am pleased to inform you that your manuscript has been deemed suitable for publication in PLOS ONE. Congratulations! Your manuscript is now with our production department. 

With kind regards,

on behalf of

Professor Nicholas J. Provart 

Academic Editor

PLOS ONE